# Mechanisms of synergy creation for social-ecological transformation: Leverage point analysis of the emergence of autonomous innovations

**Hidetomo Tajima**[1]*◉, **Tetsu Sato**[2]*◉, **John Banana Matewere**[3],
**Dorothea Agnes Rampisela**[4], **Motoko Shimagami**[5], **Shion Takemura**[6], **Mitsutaku Makino**[7]

1 Marine Fisheries Research and Development Center, Japan Fisheries Research and Education Agency, Yokohama, Japan, 2 SDGs Promotion Office, Ehime University, Matsuyama, Japan, 3 Sustainable Cape Maclear, Monkey Bay, Malawi, 4 Faculty of Agriculture and Sago Research and Development Center, Hasanuddin University, Makassar, Indonesia, 5 Institute for International Relations, Ehime University, Matsuyama, Japan, 6 Fisheries Resources Institute, Japan Fisheries Research and Education Agency, Yokohama, Japan, 7 Atmosphere and Ocean Research Institute, The University of Tokyo, Kashiwa, Japan.

◉ These authors contributed equally to this work.
* tajima_hidetomo89@fra.go.jp (HT); tetsu@chikyu.ac.jp (TS)

## Abstract

Transformation of social-ecological systems is required to overcome complex and difficult challenges and move forward toward sustainable futures for humanity and the environment. We have accumulated case examples of autonomous innovations emerging among community members in rural areas of developing countries through transdisciplinary dialogues with innovators. Collective actions among diverse actors in communities are mobilized in these autonomous innovations to promote transformations by creating synergies among resource management practices. Synergy creation has a greater impact on transformations of social-ecological systems because broader people are involved in the network, the range of covered resources and ecosystems is wider, and diverse beneficiaries are involved. We applied a novel approach to visualize leverage points in the emergence processes of synergies in autonomous innovations to understand the pathways and mechanisms of social-ecological transformation driven by these innovations. We made comparisons of 17 cases of autonomous innovations with and without synergies to clarify the mechanisms of social-ecological transformations through synergies. The process of synergy creation was complicated and several factors in the initial phases of autonomous innovations interact with each other to generate synergies. These factors included perception changes of people and innovators and visualization of new challenges. We also found that diverse functions of leverage points appearing at the initial phases played critical roles in synergy creation. Functions related to synergy creation such as the visualization of challenges, creation of values, and promoting changes of perceptions were important. Based on these analyses, we propose six guiding principles

**Data availability statement:** All relevant data are within the paper and its Supporting Information files.

**Funding:** This research was funded by the "Transdisciplinary Study of Natural Resource Management under Poverty Conditions Collaborating with Vulnerable Sectors (TDVuls) Project" under the Initiative for the Promotion of Future Earth Concept program provided by Research Institute of Science and Technology for Society (RISTEX), Japan Science and Technology Agency (JST) from 2017 to 2019 (JPMJRX16F3) and "Establishment of a Sustainable Community Development Model based on Integrated Natural Resource Management Systems in Lake Malawi National Park (IntNRMS) Project" under the Science and Technology Research Partnership for Sustainable Development (SATREPS) Program provided by the Japan Science and Technology Agency (JST) and Japan International Cooperation Agency (JICA) from 2020 to 2025 (JPMJSA1903). The funders of these projects organized review boards of the projects to conduct periodic reviews to monitor the progress of the research and provided the authors with advice and recommendations to improve research.

**Competing interests:** The authors have declared that no competing interests exist.

for local practitioners, community-based innovators, and transdisciplinary scientists collaborating with these actors to promote the transformation of social-ecological systems through the creation of synergies in autonomous innovations.

## Introduction

### Transformation of social-ecological systems through synergies

Humanity faces various serious challenges that threaten its sustainability [1–5]. Among these challenges, the degradation of natural resources and ecosystem services is the challenge that must be urgently addressed to move toward sustainable futures [6,7]. In developing countries, various socially vulnerable groups, including people living under poverty conditions, are strongly dependent on natural resources such as forests, agriculture, and fisheries [8,9]. Improving the livelihoods of these people and achieving sustainable management of natural resources is an urgent challenge, but its solution has been extremely difficult [10–12]. Social-ecological systems (SESs), in which human activities and natural and ecological systems are inextricably linked and interact in complex ways, have many uncertainties, and their behavior is extremely difficult to predict and control [13–18]. In such complex adaptive systems, wicked problems emerge that have no (or innumerable) answers and are extremely difficult to solve [19–21]. Many authors have tried to describe and propose the way for dealing with complexities and uncertainties associated with SESs, such as the cases of management of UNESCO Biosphere Reserves (BRs) in the world [22,23], and Japanese coral lagoon systems as a part of the Social-ecological Production Landscapes and Seascapes (SEPLS) in Japan and Asia [24]. To realize sustainable and equitable management of SESs and improvement of well-being of people through the integrated management of natural resources that are essential to humanity, it is important to go beyond addressing individual challenges and promote the transformation of the entire SES [9,25]. This paper aims to understand mechanisms of social-ecological transformation by analyzing the processes and factors responsible for synergy creation in autonomous innovations emerging in communities of the global south through the lens of leverage points (LPs). Autonomous innovation refers to an inclusive, bottom-up and frugal innovation consisting of a series of collective actions emerging from local practitioners with a potential to transform SESs [10,26].

People classified as socially vulnerable in various developing countries generate a variety of autonomous innovations by themselves that lead to solutions to imminent challenges [10,27–30]. In our previous studies, we found that diverse autonomous innovations emerged one after another and had a significant impact on SESs through synergies among management practices of different categories of natural resources to involve broader actors in a wide range of ecosystem services [10,30]. Transformation of SESs through synergies will lead to solving wicked problems and building sustainable futures in the process of the emergence and interaction of various autonomous innovations. Therefore, understanding the mechanisms of synergy creation in autonomous innovations will contribute to clarify the pathways of social-ecological transformation.

We focused on analyzing the mechanisms of synergy creation in autonomous innovations applying the concept of LPs as a lens to understand the mechanisms of transformation of SESs through synergies [31–36]. In complex SESs, LPs drive the dynamic emergent process of individual autonomous innovation [29]. Analyzing the factors of the emergence of synergies in autonomous innovations and the functions of LPs associated in this process are important for understanding the mechanism of transformation.

Through analyses of the emergence process of autonomous innovations by vulnerable sectors in developing countries, we found a number of autonomous innovations that achieved synergies in natural resource management practices and improved the status of different resources simultaneously [10]. Therefore, these cases can be used to analyze the specific mechanisms through which autonomous innovations transform SESs through synergies. Applying the outcome of this study to understand the mechanisms that promote synergy-based transformation of SES will help practitioners, community leaders, and relevant actors at the grassroots level to promote a series of concrete actions to solve or mitigate wicked problems emerging in complex adaptive systems.

The purpose of this study is to clarify the mechanisms by which autonomous innovations promote transformation through synergy creation by deeply analyzing the emergence processes of autonomous innovations. For this purpose, we address the following research questions.

1. What factors are important for synergy creation in the emergence processes of autonomous innovations?

2. How do leverage points function in synergy creation?

3. What should we keep in mind when promoting transformation through synergies?

## Theoretical framework

**Network analysis for leverage point identification.** Various attempts have been made to apply the concept of complex adaptive systems to understand the characteristics and behaviors of social-ecological systems to develop solutions to the difficult challenges facing humanity [15,37,38]. As an alternative to conventional top-down interventions, the significance of interaction and accumulation of collective actions to bring about system transformation has been emphasized in the theory of self-organization and emergence in complex adaptive systems. [39,40] Practical methodologies to promote SESs transformation has also been proposed, such as transition management theory. [41,42] Therefore, this paper focuses on the emergence of autonomous innovations and synergies as an important pathway of social-ecological transformation.

In our previous studies, we co-created and analyzed narratives describing the emergence processes and impacts of autonomous innovations through transdisciplinary dialogs with innovators who played a central role in creating innovations in local communities in various developing countries [10,27]. We applied the method of network analysis using causal network diagrams [43–45] to narratives describing the emergence process of autonomous innovations to identify leverage points (LPs). LPs are defined as "the part of a complex system where a small change can lead to an essential transformation of the entire system" [31]. LPs were categorized into those which were easy to intervene upon but had limited potential to bring about transformation (shallow) and those which were difficult to intervene upon but had great potential to bring about transformation (deep) [33]. While the concept of LPs has been successful in providing various guidelines for transformation in the context of policy and decision-making, there are rare examples of research aiming at realistic proposals for the effective use of LPs by practitioners who are generating concrete actions to bring about solutions [46].

In recent years, many systems thinking studies have applied network analysis methods to analyze LPs to understand the mechanisms that promote transformation of complex systems [47–53]. However, the dangers of using network analysis to identify LPs in entire complex systems have also been frequently pointed out [54–56]. Rather than focusing on the entire system, we focus on the relatively simple component of complex systems, namely, autonomous innovations,

assuming that the transformation of the entire system is promoted through synergy creation in the self-organizing processes of autonomous innovations. Therefore, the risks identified for applying network analyses to the entire system cannot be applied to extracting LPs from the emergence processes of autonomous innovations.

**Synergies to promote transformation of complex systems.** Synergy is important for promoting transformation in complex and interacting social-ecological systems. In recent years, there has been a revitalized debate on the importance of synergies, particularly between targets of Sustainable Development Goals (SDGs) and climate change responses [57–61]. For example, when natural resource management practices [58] promote improvements in the condition of several different resources through synergies, the network of people participating in the management practices is larger, the range of resources and ecosystems covered is wider, and more diverse beneficiaries occur through different supply chains [10]. Therefore, synergy creation has a greater impact on SES transformation. In addition, the improved condition of several different resources can promote the improvement of diverse aspects of human well-being [59]. For example, livelihoods will be secured and improved through the stability of diverse resources, social relations will be strengthened and expanded through the collaboration of diverse people, and health conditions will be enhanced through food security [10]. However, research on the relationship between transformation and synergy creation in SESs is at an edge. It is critically important to analyze and understand the mechanisms of synergy creation in autonomous innovations to clarify the pathways of social-ecological transformation toward sustainable futures.

## Materials and Methods

### Classification of autonomous innovations

We co-created and accumulated narratives of autonomous innovations in local communities in developing countries [see the supplementary materials of Tajima et al.2022 [10]]. In this study, we used 15 of these narratives with sufficient information for analysis. We also included two new innovations that emerged after its publication [10] to create 17 narratives for analysis. These narratives were co-created in three regions in Indonesia (Gorontalo, Polewali, and Jeneberang), one in the Philippines (Ifugao), one in Thailand (Rayon), one in Fiji (Wai), three in Malawi (Nkhotakota, Salima, and Chembe), and one in Turkey (Karapinar) (Table 1).

The numbers in parentheses on the leftmost column are the innovation numbers in Tajima et al. (2022) [10]. This paper uses the numbers originally given here (outside parentheses)

We categorized these 17 autonomous innovations into "synergistic type" (10 cases) and "single type" (7 cases) (Table 1). The synergistic type is one in which the management effects of different natural resources are generated by synergies within a single innovation. The single type is one in which management effects are obtained only for a single natural resource [10]. Natural resources include fisheries, forests, agricultural (including water) and tourism resources, and ecosystem services and landscapes that support these resources, but not cultural and social resources [10].

### Analyses of processes and factors of synergy creation

In this study, we generated causal network diagrams from narratives of the emergence process of 17 autonomous innovations [30] and compared synergistic and single types. These causal network diagrams represent the initial conditions of autonomous innovations, emergent collective actions and their outcomes, LPs, and remaining challenges as a flow along a timeline. We followed the algorithm proposed by Takemura et al.2022 [29] and drew network diagrams using our own application. To avoid potential subjectivity associated with narrative analysis, Takemura et al.2022 [29] established a structured methodology of extracting nodes and links of the causal network from narratives following the coding method proposed by Williams et al. [62] They also organized two independent teams to produce cause-and-effect lists from the narratives to identify nodes and links for network analysis. One team was in charge of extracting nodes and links of causal networks from the narratives, and the other was responsible for confirming the validity of the list of the extracted nodes and links. This formalized procedure greatly contributed to eliminating arbitrariness in narrative analysis.

**Table 1. List of 17 autonomous innovations.**

| No. | The name of innovation | Country | Area | Target resources | Type |
|---|---|---|---|---|---|
| 1 (1) | Community-based marine tourism | Indonesia | Gorontalo | Tourism, Ecosystem | Synergistic |
| 2 (2) | Improving the quality of cacao raw materials and high value-added distribution | Indonesia | Polewali | Agriculture, Ecosystem | Synergistic |
| 3 (3) | Improving cacao farm management | Indonesia | Polewali | Agriculture | Single |
| 4 (4) | Multi-species cultivation on cacao farmland | Indonesia | Polewali | Agriculture, Ecosystem | Synergistic |
| 5 (new) | Development of cacao farm tourism | Indonesia | Polewali | Tourism, Agriculture, Ecosystem | Synergistic |
| 6 (7) | Collaborative network construction | Indonesia | Jeneberang | Agriculture | Single |
| 7 (new) | Waste recycling and Tourism development | Indonesia | Jeneberang | Tourism, Landscape | Synergistic |
| 8 (8) | Improvement of rice planting method through international exchange | The Philippines | Ifugao | Agriculture | Single |
| 9 (9) | Diversification of production activities of natural rubber plantations | Thailand | Rayon | Agriculture, Ecosystem | Synergistic |
| 10 (10) | Reorganization and utilization of traditional salt making techniques | Fiji | Wai | Tourism | Single |
| 11 (11) | Small-scale aquaculture and multi-species cultivation | Malawi | Nkhotakota | Agriculture, Fisheries | Synergistic |
| 12 (13) | Seasonal fishing bans around Mbenji Island by traditional chiefs and communities | Malawi | Salima | Fisheries | Single |
| 13 (16) | Formation and operation of a tour guide association by local residents | Malawi | Chembe | Tourism | Single |
| 14 (17) | Cape Maclear Cleanup project and recycling center | Malawi | Chembe | Tourism, Landscape | Synergistic |
| 15 (18) | Organic farming by small-scale irrigation linked to educational activities | Malawi | Chembe | Agriculture, Tourism | Synergistic |
| 16 (19) | Efforts by fishers to create satoumi-type fishing grounds | Malawi | Chembe | Fisheries, Ecosystem | Synergistic |
| 17 (20) | Cultivation and sale of pickled salad melons requiring small amount of irrigation water | Turkey | Karapinar | Agriculture | Single |

In this study, we define "autonomous innovation" as "a state in which multiple collective actions (see below) emerge in response to challenges that are manifested in initial conditions, and feedback loops in a network diagram are formed from the results of these actions to dynamically move the process toward solutions to the challenges". The formation of feedback loops implies a state in which resolution or improvement is achieved with respect to the challenges manifested in the initial conditions. Thus, a causal network diagram is a snapshot of a specific point in time. We define "collective action" as " a part of a network diagram from a node indicating the initiation of some changes in perceptions and/or activities with respect to the management of a specific resource to a node indicating the generation of certain outcomes related to the relevant resource." Fig 1 shows examples of network diagrams of synergistic and single types representing collective actions consisting of autonomous innovations.

With these network analyses, we identified three types of LPs proposed by Takemura et al. 2022. [29] LPs were defined as a single node in a causal network and classified into three types: LP(in), LP(out), and LP(all). LP(in) functioned to integrate new components into the emergence processes of autonomous innovations, LP(out) played a role in creating new components in the emergence processes, and LP(all) supported the entire processes through the integration and

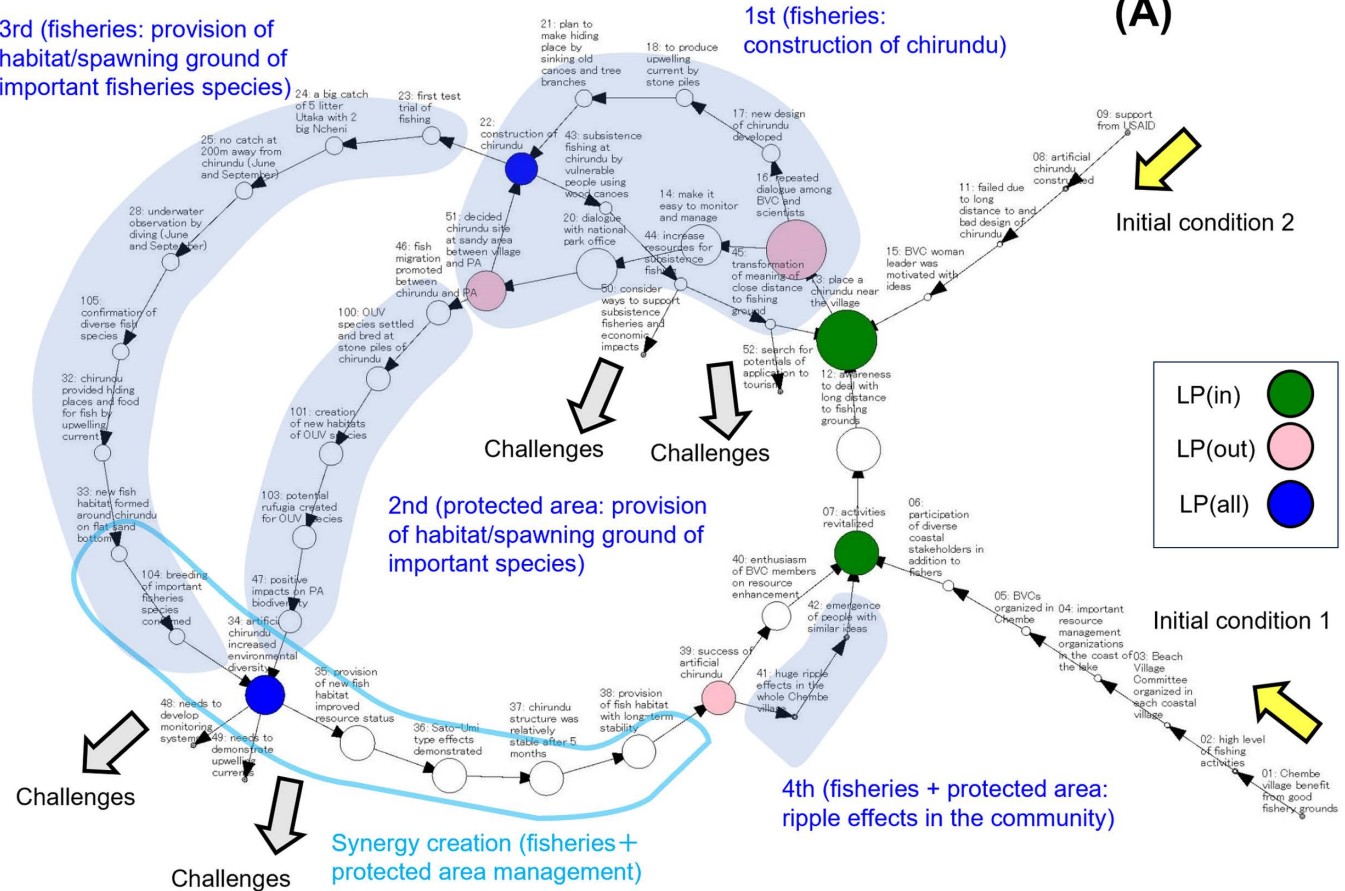

**Fig 1. Network diagrams of synergistic types (A) and single types(B).** The yellow arrows represent the initial conditions for autonomous innovations, the blue shaded areas represent individual collective actions, the blue letters represent the target resources and content of collective actions, and the gray arrows represent the remaining challenges. The three types of LPs are represented as green, pink, and blue nodes. In the synergistic type, synergies are represented as one or more nodes surrounded by light blue lines. See text for details. **(A)** A synergistic type (No. 16) "Efforts by fishers to create satoumi-type fishing grounds." Synergy has been created by the emergence of collective actions for fisheries resource and ecosystem management. **(B)** A single type (No.6), "Collaborative network construction." Three collective actions related to agricultural resource management have emerged.

creation of feedback loops in the networks. Using this definition of LPs, we attempted to identify the mechanisms by which LPs promote the transformation of SESs through the emergence of autonomous innovations [30]. We found that in the emergence process of individual autonomous innovations, LPs performed diverse functions, and a single LP sometimes performed multiple functions [30]. These functions of LPs are analyzed in detail as described below to identify important enablers of SES transformation.

All autonomous innovations involve multiple collective actions. Since the first collective action of an autonomous innovation was related to a natural resource management challenge that the innovator was most concerned about, we named it "First action". For example, in Fig 1A, the first action is "construction of chirundu (artificial fishing reef)". In this case, "idea to place a chirundu near the village" (node no.13) played an important role in initiating collective action. This node is LP(in) playing a role to integrate past experiences and new ideas to accelerate first collective action. In addition, "repeated dialog among BVC, knowledge translators and scientists" (no.16: LP(out)) reinforced the action. The nodes "chirundu site decided on sand bottom between village and PA (no.51: LP(out))" and "construction of chirundu (no.22: LP(all))" facilitated next collective actions. In Fig 1B, the first action was "Construction of tertiary channels at the most downstream

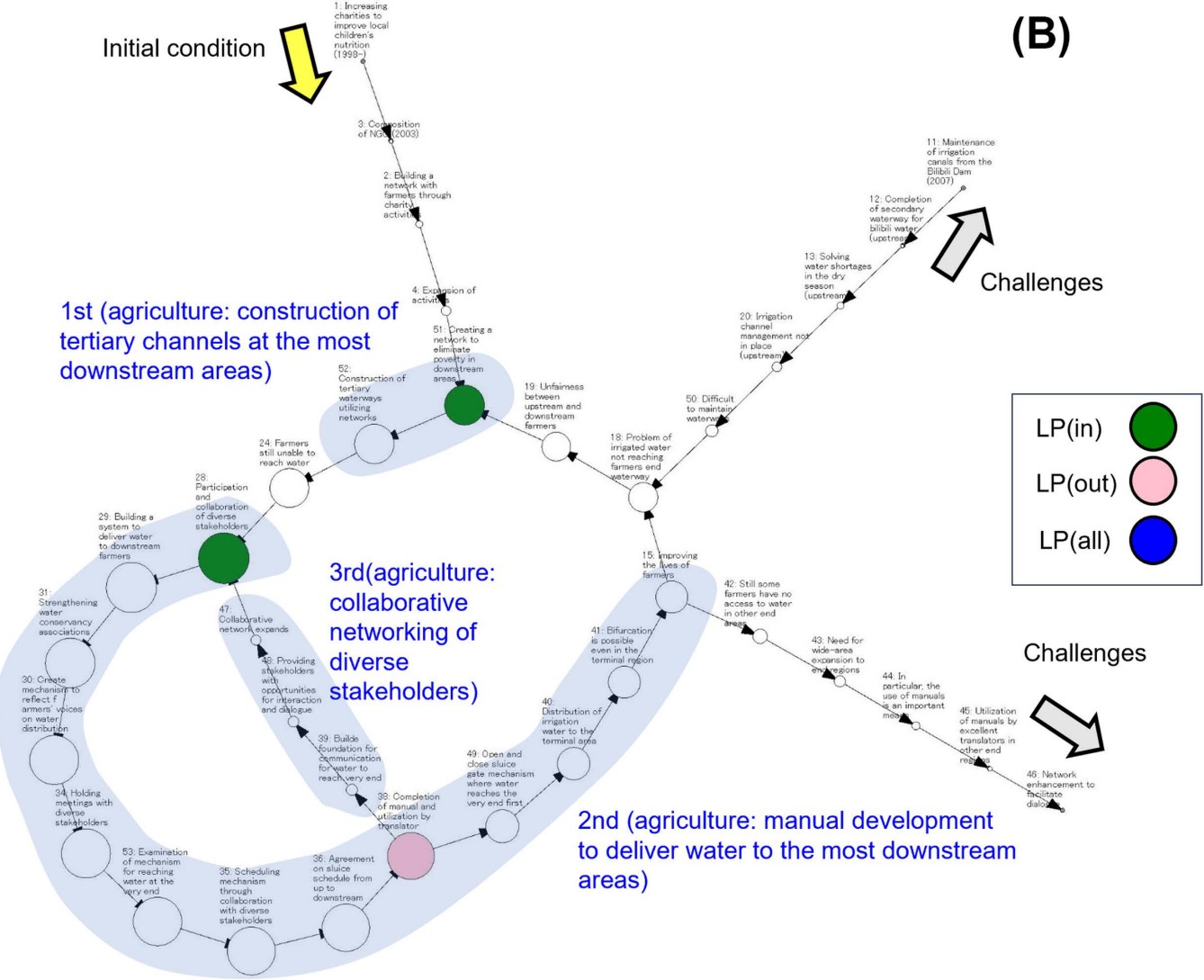

**Fig 1.** Continued.

areas" which stimulated the awareness and mutual understanding of actors and the node "Creating a network to eliminate poverty in downstream areas (no.51: LP(in))" stimulated the emergence of subsequent collective actions by integrating initial conditions. We refer to a series of collective actions that emerge following the first action as "Second action", "Third action", etc. (Fig 1). For example, in Fig 1B, the second action is "manual development to deliver water to the most downstream areas" and the third action is, "collaborative networking of diverse stakeholders."

We define "synergy" as "a state in which autonomous innovations have a positive impact on more than one resource in addition to the original target resource. " For example, as shown in Fig 1A, this autonomous innovation impacts fisheries resource management, as artificial fishing reefs provide effective fishing sites and breeding grounds for important fish species. Furthermore, it has a synergistic impact on ecosystem management by providing habitats and breeding grounds for important fish species for the management of a World Natural Heritage site.

We hypothesized that in a synergistic manner, the outcome of first collective action targeting the highest priority challenges for innovators (e.g., fisheries resource enhancement in Fig 1A) would facilitate the emergence or enhancement of

second collective action targeting different resources (e.g., effects on ecosystem management in Fig 1A). Thus, if collective actions related to multiple resources have produced outcomes at a particular point in the time series, this indicates the emergence of synergies. The factors responsible for synergy creation should be identified in the process before this point. We examined the initial conditions and content of individual collective actions of the synergistic types to determine the part of the network diagrams where synergies emerged. We searched for factors that influenced synergy creation in the process before this part. We also examined the network diagrams of single types to determine if these factors responsible for synergy creation were also present.

Because LPs perform different functions for the emergence of autonomous innovations [30], we extracted and classified the functions of LPs based on the contents of the nodes before and after the LPs in the network diagram in synergistic and single types. We also compared the total number of LPs and the number of each type of LPs in each innovation. We also analyzed the positions in network diagrams where the three types of LPs emerged.

Based on the results of these analyses combined with implications from our previous papers [10,29,30], we propose the guiding principles of promoting transformation through synergies, which are addressed to potential innovators playing a central role in the emergence of autonomous innovations, as well as to actors such as transdisciplinary scientists and local leaders who work with these innovators.

## Results

In the synergistic type innovations, the first and second actions targeted different resources in all 10 cases (e.g., Fig 1A) in contrast to the single type (e.g., Fig 1B). In six synergistic types, the first action was followed by the emergence of the second action. In the remaining four cases, the first and second actions emerged almost simultaneously and in parallel. In these cases, we assigned the first action to one related to the more important challenges for innovators, judged from the initial conditions. In all 10 synergistic cases, synergies occurred before or after the conclusion of the second action (e.g., Fig 1A). The only difference between cases in which the first and second actions emerged consecutively and concurrently was the timing of the start of the second action. Synergies commonly occurred at the end of the second action. Therefore, we did not distinguish between these two types in our later analysis. Synergies emerged when the outcomes of the first actions promoted the emergence or reinforcement of the second action. Therefore, the phase from the initial condition to the transition period from the first action to the second action plays a decisive role in synergy creation.

Fig 2 presents an example of a network diagram illustrating the key factors; perception changes and visualization of new challenges, involved in the creation of synergies and social-ecological transformation. Under the initial condition of a decline in fisheries production and limited livelihood opportunities, the first action, "start of diving tourism and increase of tourists", emerged. The outcome of the first action has made visible new challenges related to management practices for different resources (ecosystem management). People's perceptions changed (enhancing conservation mindsets) during the second action "taking actions to conserve coral reefs". Synergies (tourism + ecosystem management) emerged after the completion of the second action to promote conservation-oriented tourism. The third action, "promotion of tourism development," was dynamically activated to transform the social-ecological system through synergy creation to form a new feedback loop to promote the process of managing multiple resources. Future challenges remain regarding collaboration with broader stakeholders to develop tourism industries and conserve coral reefs.

We found no significant difference in the total number of LPs and the number of LPs per type that emerged during the entire autonomous innovation process between synergistic and single types. We classified the functions of each LP appearing from the initial condition to the period of transition from the first action to the second action (Table 2), which were judged by the contents of nodes before and after each LP in the network diagram. The number of LP functions per innovation was significantly larger for the synergistic type than for the single type (Fig 3). LPs in the synergistic and single types shared many functions except for the functions related to synergy creation (Table 2). These LP functions are particularly important enablers for synergy creation.

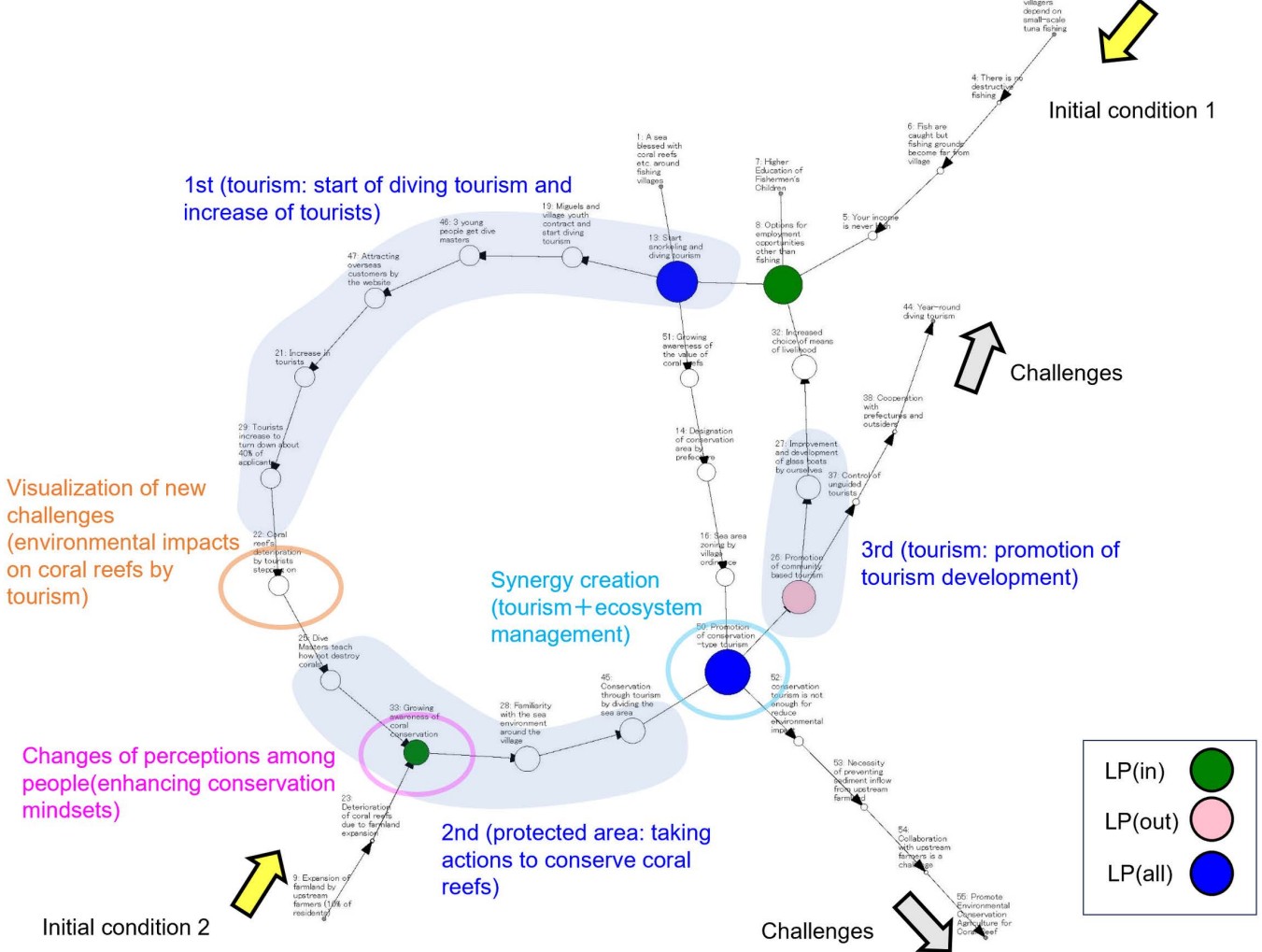

**Fig 2. Example of a network diagram showing the process of synergy creation and the responsible factors.** Network diagram of No.1 Community-based marine tourism, showing the process and factors of synergy creation in autonomous innovation. Orange and pink indicate the key factors that generate synergies (visualization of new challenges, changes of perceptions among people). The other symbols and labels are the same as those in Fig 1A.

Throughout the entire process of autonomous innovations, we found cases with challenges that might lead to the creation of new synergies in the future [10]. In six of the 10 synergistic types (Nos. 1, 2, 4, 7, 15, 16 of Table 1), the visualization of challenges with potential for future synergies was found in various phases of the emergence processes of autonomous innovations (e.g., Fig 4). In the two synergistic cases of Polewari, Indonesia (Nos. 2 and 4 of Table 1, also see Fig 4), the same new challenge, "possibility of cacao farm tourism collaborating with consumers," was visualized through a change of perception of innovators, which served as the initial conditions for another autonomous innovation (No.5 of Table 1) to create synergies. In the single type, challenges that led to future synergies were not observed.

As mentioned above, some LPs were found to have the function of visualizing challenges that could lead to new synergies. Therefore, we confirmed the existence of LPs that have the function of visualizing challenges to lead to the creation

**Table 2. Number of LP functions observed before the end of the first action in synergy and single types.**

| Functions of the LPs | Synergistic type (10) | Single type (7) |
|---|---|---|
| Emergence of collective action | 11(9) | 5(5) |
| Strengthening the foundation for collective action | 5(5) | 1(1) |
| Collective action development | 2(2) | 1(1) |
| Creation of values | 2(2) | 3(3) |
| Creation of values leading to synergies | 2(2) | 0(0) |
| Visualization of motives | 5(5) | 3(3) |
| Visualization of new challenges | 4(3) | 4(3) |
| Visualization of new challenges leading to synergies | 2(2) | 0(0) |
| Perception changes of innovators | 5(3) | 0(0) |
| Perception changes of people | 2(2) | 0(0) |
| Total | 40 functions/29 LPs | 17 functions/14 LPs |

Numbers in parentheses represent the number of innovations with relevant LP functions. Shaded cells indicate LP functions appearing only in synergistic types.

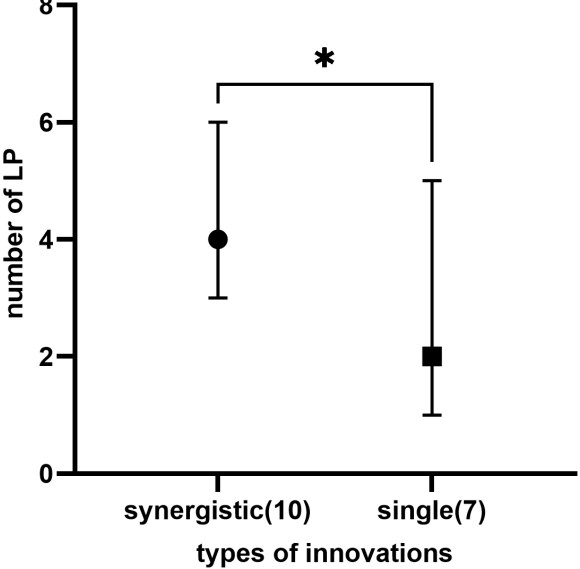

**Fig 3. Difference in the number of LP functions observed in autonomous innovations between synergistic and single type.** The median and 95% confidence intervals are represented. Numbers in parentheses represent the number of innovations. (Mann-Whitney test p = 0.0260).

of new synergies (e.g., LP(out): no.35 in Fig 4). We found such LPs in five of the six cases (No.1, 2, 4, 7, 15 of Table 1). Three LPs were of LP(all) and two were of LP(out). LP(all) and LP(out) seemed to be important for visualizing challenges leading to new synergies in the future.

## Discussion

Studies have accumulated regarding the complexities of social-ecological systems and difficulties to bring solutions for the wicked problems associated with them [14,18,20]. Many authors also indicated the importance of going beyond working on individual challenges in isolation to address the transformation of the whole system [35,63]. However, we found limited

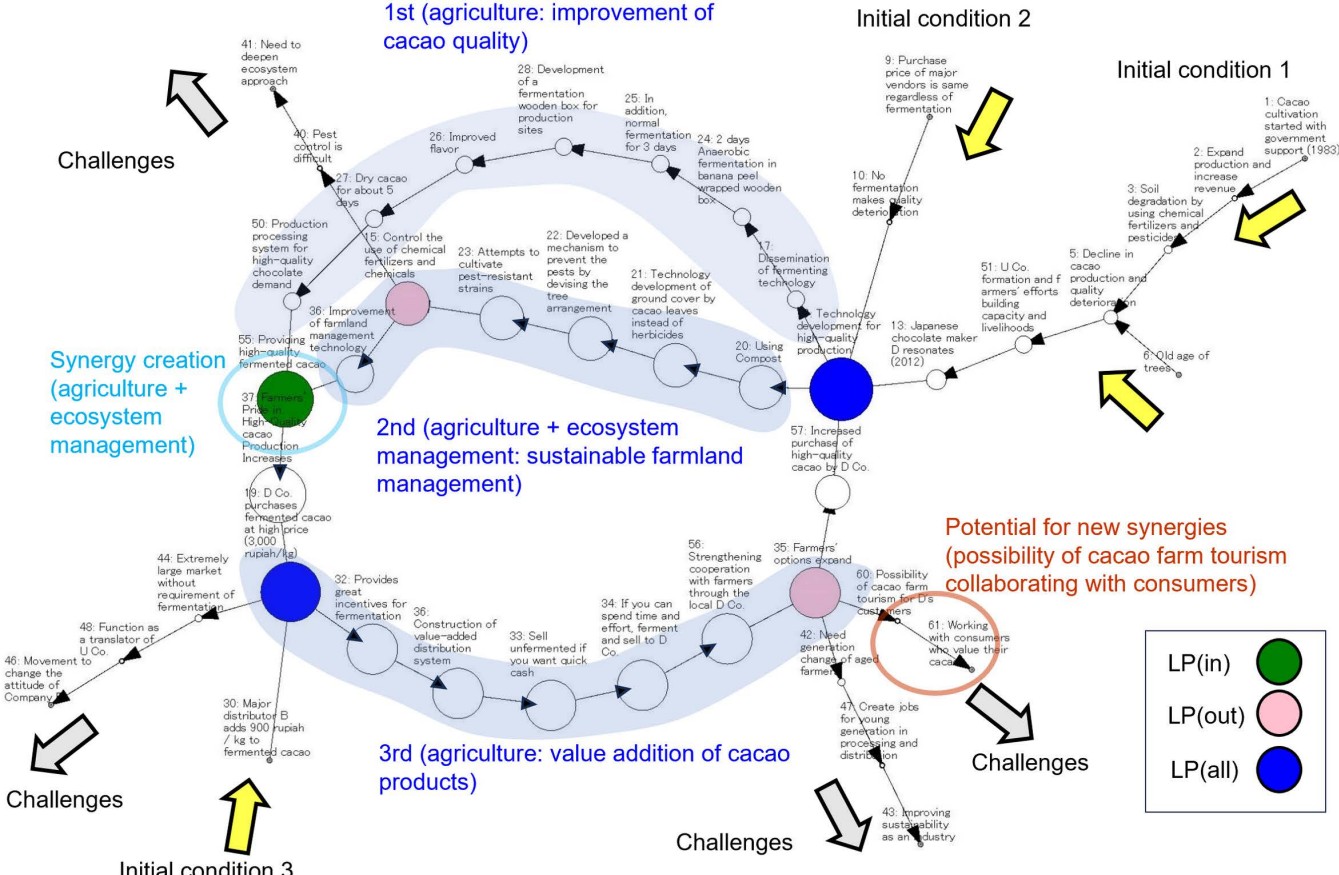

**Fig 4. Visualization of challenges leading to new synergies in the future.** An autonomous innovation that visualizes challenges leading to new synergy creation in the future (No.2 Improving the quality of cacao raw materials and high value-added distribution). A new challenge was visualized at the end of the third action after the synergy (brown: "possibility of cacao farm tourism collaborating with consumers"). In this case, node no.35 ("Farmers' options expanded") served the function of visualizing the challenges leading to future synergies. The symbols and labels in this figure are the same as in Fig 1.

examples of studies proposing tangible mechanisms of transformation for diverse societal actors to apply in their practices and actions [64]. We believe that understanding and applying the factors of social-ecological transformations elucidated by leverage point analyses of the emergence of autonomous innovations and synergies in this study have significant academic and societal impacts [29,30].

Fig 5 shows a schematic of the factors identified as leading to the synergies by comparing the 10 synergistic types with the seven single types. In the six synergistic types, innovators were conscious of synergy emergence from the early phase ("Before 1st action"; the gray shaded area in Fig 5). The types of LPs that emerged before the first action were all LP(in), whether synergistic or single. In many cases, the LP(in) strengthened the basis for collective action. In the four cases of the synergistic type, innovators changed their perceptions that the challenges they were trying to tackle were also related to different resources and that synergies could promote the resolution of these multiple challenges to mobilize transformation of SESs [58]. In all synergistic types, one or more of these three factors were functioning in the "Before 1st action" phase (Fig 5). On the other hand, in the single type, there was one case in which innovators were aware of the synergy potential and one case in which the LP(in) strengthened the foundation of the activity. Therefore, although these factors are important for synergy creation, it cannot be concluded that synergy is determined in "Before 1st action" phase.

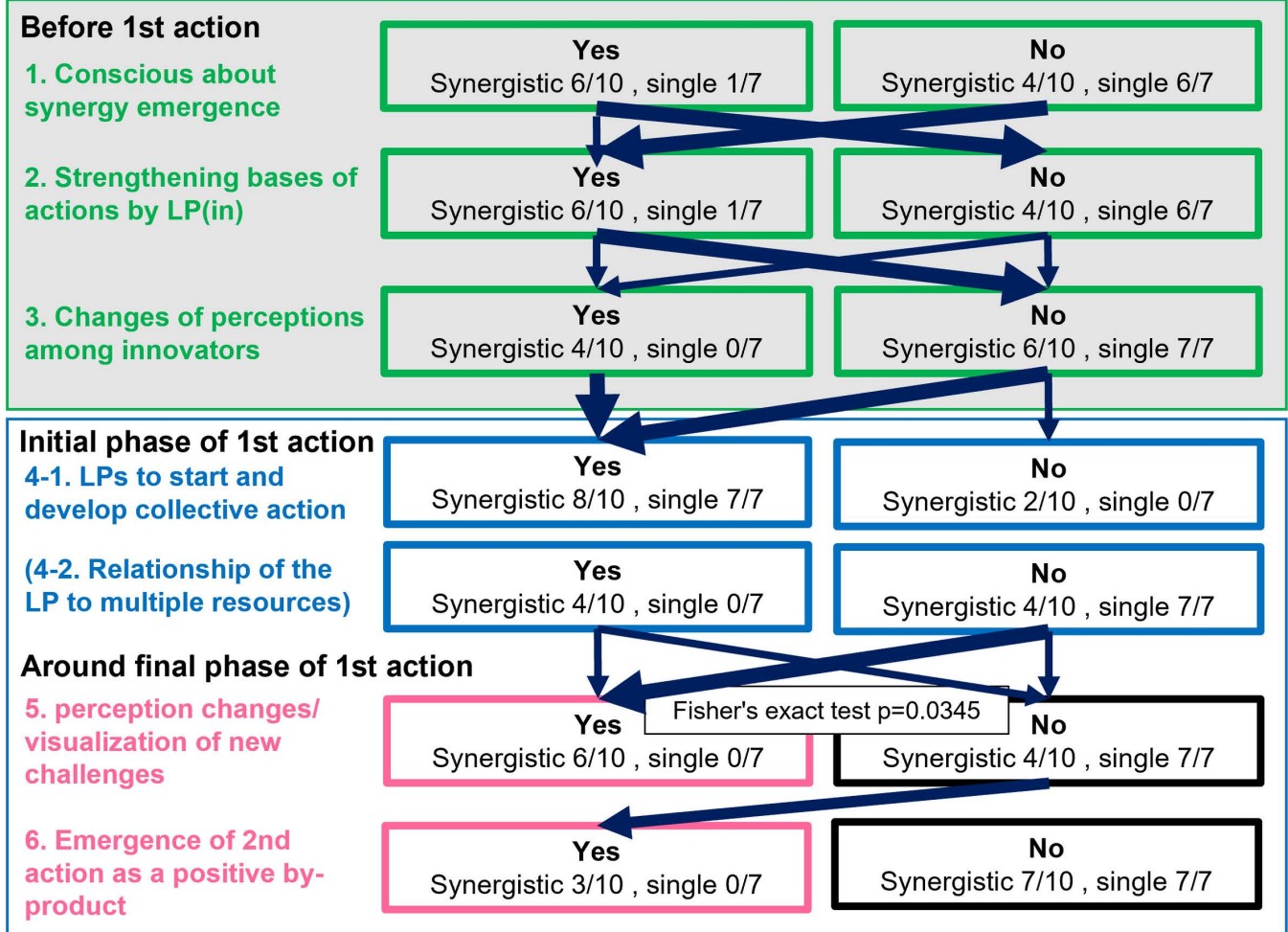

**Fig 5. Factors of synergy creation in the early stage of autonomous innovations.** The gray shaded area represents the phase before the start of the first action, and the white background area represents the period from the initial to the final phases of the first action. The columns on the far left represent the factors identified in the network analyses that are considered important in creating synergies. The numbers assigned to each factor are for convenience only and do not represent a time series or causal relationship. The numbers in the boxes indicate the number of cases where each factor was found in synergistic or single types (number of cases/parameter). The thickness of the arrows is proportional to the number of cases that followed each path for the synergistic types. The thicker the arrow, the more cases followed the path in the synergistic type. See text for details.

In the "Initial phase of the first action", three types of LPs emerged, whether synergistic or single, to activate and develop collective action (Fig 5, "LPs to start and develop collective action"). In half (4/8) of the synergistic types, LPs were working to activate and develop collective action across multiple resources. In contrast, no such LPs were observed for the single types (Fig 5, "Relationship of the LP to multiple resources"). Therefore, the function of LPs related to multiple resources played an important role as the deep LPs to promote synergy to bring about transformation [33].

In the synergistic type, the outcomes of the first actions resulted in innovators and people changing their perceptions and/or visualizing new challenges (Fig 5, "perception changes/visualization of new challenges"). However, these factors were not observed in all single types. Second actions targeting different resources emerged in six cases of synergistic type (Nos. 1, 4, 7, 11, 14, 15 of Table 1). We considered the changes of perceptions among people and the visualization of new challenges based on the results of the first actions to be important factors in the emergence of synergies. In the

remaining three synergistic cases (Nos. 2, 9, and 16 of Table 1), people did not change their perceptions, nor were the new challenges visualized. In these three cases, a second action emerged as a positive by-product of the first action (Fig 5, "Emergence of 2nd action as a positive by-product"). The remaining case (No. 4 of Table 1) was a case in which the innovator was strongly aware of the synergy and changed his/her perception during the "Before 1st action" phase to design collective actions to realize synergy in the "Initial phase of the first action". However, this was a rare case.

Based on the results of the above analysis, we conclude that synergy creation is determined when the first action results are realized. Our approach to apply network analyses on the emergence of autonomous innovations, a relatively simple component of the complex system, was successful to identify important factors of synergy creation to facilitate self-organizing processes [60]. Synergies are promoted by various factors in the process until the stage of the outcome generation of the first action, indicating the complexity of the pathways of synergy creation. The pathway to synergy is not deterministic, but at least perception changes, visualization of new challenges, and emergence of positive by-products play important roles. Furthermore, only autonomous innovations that have already generated synergies were found to have the potential to create new synergies in the future, opening the pathway toward adaptive and dynamic processes of transformation [29]. Therefore, the factors of synergy generation identified in this study are important for creating synergies and transforming SESs in the future.

## Conclusion, Limitations, and Future Research Agenda

### Conclusion and guiding principles

Synergies are considered to strengthen societal impacts and effectively promote transformation because they expand the network of people, influence a wider range of resources and ecosystems, and involve a variety of beneficiaries. In this study, we found that synergy creation was determined before the first action outcomes appeared. In autonomous innovations without synergies, the challenges leading to new synergies were not observed. To promote the transformation of social-ecological systems through synergies in autonomous innovations, it is important to be aware that the early stages of innovations are particularly important. If synergies cannot be created in the early stages of autonomous innovations, synergies will not emerge even in the future, and the transformation of SES through synergies will be unlikely. The tendency to focus on specific resource management practices of interest, as seen in single type autonomous innovations, may undermine the potential for synergies.

In the initial conditions of autonomous innovations, we found that the innovator's awareness of synergy creation, the functioning of LP(in) to strengthen the basis for collective actions, and the innovator's perception changes on the importance of managing multiple resources were critical factors in synergy creation. At least one or more of these factors contributed to the emergence of synergies. Therefore, it is important for innovators to recognize the potential for synergy creation at the very early stages of autonomous innovations and strengthen the basis of their collective actions. However, this alone does not determine the emergence of synergies because even single type innovations involved at least a few of these factors.

In many cases, when the outcomes of the first collective action emerged, second actions targeting different resources were initiated through changes in people's perceptions and the visualization of challenges leading to the creation of new synergies. Sato et al. [9] proposed five enablers of social-ecological transformation. Among them, the enabler of "effective knowledge translation" is considered to promote changes in people's perceptions. "Identify and visualize values" based on the outcomes of the first actions are also effective. In eight of the 10 cases (Nos. 1, 2, 4, 5, 7, 9, 14, 16 of Table 1), visualization of new values and motivations were related to public values such as ecosystem and landscape management. We also noted that in four cases (Nos. 1, 2, 4, 14 of Table 1), first collective actions aimed at resolving trade-offs between resource use and supporting services of ecosystems and landscapes led to synergies. These are considered to correspond to enablers of social-ecological transformations, including the "visualization of new values" and the "provision of

options and opportunities" [9,30]. In addition, changes in people's perceptions leading to synergy may function as a deep LP, which is difficult to intervene but is considered to have a significant impact on transformation [33,35,65].

Throughout synergy creation processes, many innovators were aware of the importance of improving diverse aspects of human well-being, such as ensuring resource stability through ecosystem and landscape management, strengthening collaboration with people, and improving livelihoods through resource management [30]. The improvement of diverse aspects of human well-being will contribute significantly to changes in perception and visualization of challenges as deep LPs.

Even when people did not change their perceptions or new challenges were not visualized, second actions targeting different resources emerged as a positive by-product of the first action in three cases. Therefore, it is also important to be aware of the potential of collective actions to produce impacts on different resources as positive by-products from the very beginning of autonomous innovations [66,67].

Analyzing synergy creation processes through the lens of LPs has greatly advanced our understanding of the mechanisms of social-ecological transformations. Various types of LPs that promote collective actions across multiple resources in the early stages of the first action may contribute to synergy creation. Since LP(all) and LP(out) have a function to generate new links [29], they may play a significant role in visualizing challenges related to the emergence of new synergies. The three types of LPs play diverse functions in all synergy creation processes and are considered important both for synergies that have already emerged and for creating the possibility of new synergies in the future. In particular, functions such as visualization of challenges leading to synergies, creation of values leading to synergies, and promoting changes of perceptions are important.

Based on these results and discussions, we propose the following six guiding principles to promote the transformation of social-ecological systems through the creation of synergies in autonomous innovations.

- Strengthen the foundation of collective actions based on awareness of potential synergies in the early stages of autonomous innovations.

- Design and implement collective actions deliberately to generate synergies as positive by-products.

- Be aware of the importance of the function of LPs in promoting changes in people's perceptions and the visualization of new values and challenges.

- Not to be satisfied with the results of collective actions aimed at solving urgent challenges but to deepen the motivation for different resource management practices.

- Promote the emergence of collective actions involving the enhancement of public values such as ecosystems and landscapes.

- Design and implement collective actions to improve diverse aspects of human well-being.

By taking these guiding principles into consideration and encouraging the emergence of collective actions to create synergies through autonomous innovations, local practitioners, community-based innovators, and transdisciplinary scientists collaborating with these actors can move one step forward toward solving the wicked problems that humanity faces. Policymakers and development aid agencies should consider these guiding principles to support grassroots practices to mobilize the emergence of autonomous innovations with synergies.

## Limitations and Future Research Agenda

Social-ecological systems are extremely complex and continue to change dynamically beyond our imagination. Unprecedented and difficult challenges arise from one after another, and there is no end in sight for the efforts to find solutions. More frequent and timely production of knowledge about the dynamism of SESs is required to address this limitation. The network

analysis in this study is based on the transdisciplinary co-creation of narratives among community-based innovators and scientists [10]. The co-creation process involves repeated and time-consuming dialogues of all participants based on mutual trust. More frequent dialogues will certainly improve the time resolution of the narratives as a snapshot at a particular point in time. The methodologies of transdisciplinary co-creation of narratives proposed in our previous studies [10,68] should be further refined to allow frequent production of narratives to analyze effective pathways toward sustainable futures at appropriate timing. This can be achieved by deepening mutual respect and trust and improving approaches to promote dialogues with the use of cutting-edge techniques, including boundary objects [69,70]. We will explore the potential of network diagrams indicating leverage points as novel and effective boundary objects [29]. We further endeavor to develop an AI-assisted application to conduct rough extraction of nodes and links from the narratives, thereby minimizing the required time for the narrative analyses to create a series of network diagrams as snapshots with shorter time intervals to capture the dynamic changes of SESs.

The transdisciplinary approach to trust-based dialogues is a powerful tool for reducing the arbitrariness associated with narrative analyses. All contents of the narratives, the series of nodes and links extracted from the narratives and the network diagrams representing leverage points should be created and confirmed together with the innovators responsible for the emergence of autonomous innovations to secure reproducibility. In the analysis of this study, we recognized that a certain arbitrariness remained during the extraction of nodes and links and the characterization of leverage point functions. We made our best efforts to minimize this arbitrariness by applying the consistent procedures proposed by Takemura et al. [29]. The improvement of transdisciplinary methodologies for trust-based dialogues as proposed above will certainly minimize arbitrariness throughout all analytical procedures by continuous co-production of knowledge among community-based innovators and scientists. In addition, AI-assisted applications for LP analyses may help to conduct more robust analyses with higher reproducibility, both for extraction of nodes and links and characterization of LP functions.

In this study, we focused on the synergy creation between the management practices of different types of natural resources, because they were especially important for livelihood and well-being of socially vulnerable sectors in the global south [8,9]. Farmers and fishers, as well as small-scale processors, traders, and tour operators were assumed to be the main beneficiaries of the success of integrated and sustainable management of these resources to mitigate wicked problems. However, in order to address profound impacts of synergies upon the social-ecological transformation, we should move one step further to analyze synergy creation mechanisms among broader societal sectors working to improve human-nature interactions. These sectors include education, health, social welfare, and social enterprises [71,72]. A knowledge gap exists to identify synergy creation factors across natural resource users/managers and such a broad range of actors, waiting for proactive and creative research to generate synergies for transformation toward just, equitable and sustainable futures.

This study conducted a causal network and leverage point analysis focusing on the mechanism of synergy creation in the emergence of autonomous innovations in developing countries. As a result, we have clarified the factors of synergy emergence and the functions of LPs to propose guiding principles to promote SES transformation through autonomous innovations, which are certainly applicable to rural communities in the global south. It will be of great significance to accumulate narratives of diverse autonomous innovations in different contexts and explore the mechanisms of social-ecological transformations applicable to diverse regions and communities around the world, including communities in highly industrialized countries. Further research on the mechanisms of synergy creation based on various concrete examples from around the world with heterogeneous contexts can contribute to the global promotion of social-ecological transformation and the realization of sustainable futures.

## Supporting information

**S1. Causal network diagrams of 17 autonomous innovations.pdf.** Initial conditions, collective actions, remaining challenges, leverage points, timing of synergy creation, factors of synergy creation, and potential for new synergies are presented for each autonomous innovation. These diagrams were used for the analysis to develop Fig 5.
(PDF)

**S2. Causal relationships of 17 autonomous innovations.pdf** The nodes and links for causal network analyses are represented for each autonomous innovation.
(PDF)

**S3. Centrality values of all nodes of causal networks.pdf.** The betweenness centrality and leverage centrality values for all nodes of causal networks are summarized and LPs are indicated for each autonomous innovation.
(PDF)

**S4. Number of LP functions for Fig 3.pdf.** The data for analysis to draw Fig 3 is represented.
(PDF)

## Acknowledgments

This paper benefited greatly from discussions with the members of two consecutive transdisciplinary research projects (Transdisciplinary Study of Natural Resource Management under Poverty Conditions Collaborating with Vulnerable Sectors "TD-Vuls", and Establishment of a Sustainable Community Development Model based on Integrated Natural Resource Management Systems in Lake Malawi National Park "IntNRMS". We are grateful to community-based innovators as partners of this research, including Yunis Amu, Guntur Amu, Herwin Hartawan Soekirman, Amrisal Alamzah Sennang, Umar, Untuk Indonesia Hijau, Dari K, Co. Ltd., Adam, Idrus, NGO Pelangi, Kamaruddin Rola, Hasanuddin Nassa, Ilyas Laja (Indonesia), Rolando Addug (Philippines), Kitsana Sakpeerakul, Kittisak Sakpeerakul (Thailand), Anareta Dokai (Fiji), Arnold Rumaka, Chief Makanjira, Lackson Maliwanda, Cape Maclear Tour Guide Association, Brighton Ndawala, Sinthana project, Zaret Kalanda, Madothi Beach Village Committee (Malawi), and Hayri Merdane (Turkey). Atsuko Fukushima and Kyoko Jomae provided administrative support throughout the research.

## Author contributions

**Conceptualization:** Hidetomo Tajima, Tetsu Sato, Shion Takemura.

**Data curation:** Hidetomo Tajima, Tetsu Sato, Shion Takemura.

**Formal analysis:** Hidetomo Tajima, Tetsu Sato, Shion Takemura.

**Funding acquisition:** Tetsu Sato.

**Investigation:** Hidetomo Tajima, Tetsu Sato, John Banana Matewere, Dorothea Agnes Rampisela, Motoko Shimagami.

**Methodology:** Hidetomo Tajima, Tetsu Sato, Shion Takemura, Mitsutaku Makino.

**Project administration:** Tetsu Sato.

**Resources:** John Banana Matewere, Dorothea Agnes Rampisela, Motoko Shimagami.

**Software:** Hidetomo Tajima, Tetsu Sato, Shion Takemura.

**Supervision:** Hidetomo Tajima, Tetsu Sato, Mitsutaku Makino.

**Validation:** Hidetomo Tajima, Tetsu Sato, John Banana Matewere, Dorothea Agnes Rampisela, Motoko Shimagami.

**Visualization:** Hidetomo Tajima, Tetsu Sato, Shion Takemura.

**Writing – original draft:** Hidetomo Tajima, Tetsu Sato.

**Writing – review & editing:** Hidetomo Tajima, Tetsu Sato, John Banana Matewere, Dorothea Agnes Rampisela, Motoko Shimagami, Shion Takemura, Mitsutaku Makino.

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
