## [Decision Letter · Decision Letter 0]

26 Dec 2024

PONE-D-24-53688Mechanisms of synergy creation for social-ecological transformation:  Leverage point analysis of the emergence of autonomous innovationsPLOS ONE

Dear Dr. Tajima,

Thank you for submitting your manuscript to PLOS ONE. After careful consideration, we feel that it has merit but does not fully meet PLOS ONE’s publication criteria as it currently stands. Therefore, we invite you to submit a revised version of the manuscript that addresses the points raised during the review process.

We look forward to receiving your revised manuscript.

Kind regards,

Enrico Ivaldi

Academic Editor

PLOS ONE

Journal Requirements:

2. Thank you for stating the following financial disclosure: This research was funded by the “Transdisciplinary Study of Natural Resource Management under Poverty Conditions Collaborating with Vulnerable Sectors (TD-Vuls) Project” under the Initiative for the Promotion of Future Earth Concept program provided by Research Institute of Science and Technology for Society (RISTEX), Japan Science and Technology Agency (JST) from 2017 to 2019 (JPMJRX16F3) and “Establishment of a Sustainable Community Development Model based on Integrated Natural Resource Management Systems in Lake Malawi National Park (IntNRMS) Project” under the Science and Technology Research Partnership for Sustainable Development (SATREPS) program provided by the Japan Science and Technology Agency (JST) and Japan International Cooperation Agency (JICA) from 2020 to 2024 (JPMJSA1903).

Reviewers' comments:

Reviewer's Responses to Questions

**Comments to the Author**

1. Is the manuscript technically sound, and do the data support the conclusions?

Reviewer #1: Partly

Reviewer #2: No

2. Has the statistical analysis been performed appropriately and rigorously? 

Reviewer #1: Yes

Reviewer #2: Yes

3. Have the authors made all data underlying the findings in their manuscript fully available?

Reviewer #1: Yes

Reviewer #2: Yes

4. Is the manuscript presented in an intelligible fashion and written in standard English?

Reviewer #1: No

Reviewer #2: Yes

5. Review Comments to the Author

Reviewer #1: Dear Author(s),

Thank you for giving me the possibility to read and evaluate your work. The article covers an interesting and extremely timely topic, exploring the social and ecological transformation of systems and related challenges related to today's uncertainty, that is coherent with the focus of Plos One journal.

Therefore, the manuscript covers an extremely up-to-date and interesting topic. Despite the interest in the topic, I believe the article can be further enriched, and I hope my advice and suggestions can help the author(s) in strengthening their manuscript.

- Introduction:

The introduction sections in my opinion are too fragmented and reduce the readability of the article. I suggest the authors keep section 1, i.e., introduction, but without providing sub-sections for individual aspects. As for the introduction section, I suggest checking and reinforcing a few crucial elements that must be present in the introduction of a paper namely (1) the purpose of the study; (2) the relevance and significance of the study; (3) how it seeks to answer the set research question; and (4) the main contributions of the article.

In my opinion, some of the content included in the introduction could be channeled into a section 2 called for example literature review or theoretical framework as it is more specific and focused on more detail, compared to the initial preamble. Particularly in that the theoretical lens through which the topic is to be analyzed is specified.

Therefore, I would advise the authors to restructure pages 3-5 by constructing an introduction with the purpose and research question and then a second, more theoretical section explaining in detail the theoretical lens chosen for the article and consequently renumber the sections to follow.

- Materials and Methods:

I suggest the authors place the Table within the text closely so that the text can be read on page 6. I also suggest the authors avoid excessive fragmentation in the second session however by providing a subdivision only for data collection and implemented data analysis.

- Results and Discussion:

Also, for this section, I would avoid excessive fragmentation of the text and provide for splitting the Results and Discussion sections. I would advise the authors to enrich a bit their results, which appear a bit too descriptive and fail to represent more qualitatively the results that emerged also considering the theory and potential implications. This part could go to enrich an individual section based on the Discussion of Emerging Findings. Enhancing what may be the implications for society and policymakers. I would enrich this concluding part by providing a section on “Conclusion, Limitations and Future Research Agenda” precisely to highlight the limitations of the study and how these can be overcome by future research.

I hope you will find these comments and suggestions helpful in further developing this study and I wish you all the best for developing your work!

Reviewer #2: The manuscript titled “Mechanisms of synergy creation for social-ecological transformation” explores an innovative and relevant topic by analyzing mechanisms for creating synergies in socio-ecological transformations through leverage points. This approach is particularly effective for studying complex adaptive systems. It employs a well-justified and clearly explained methodology, using causal network diagrams to analyze processes of autonomous innovation. This technique allows for precise mapping of key factors and interactions. Moreover, the article provides six practical guiding principles for fostering socio-ecological system transformations, designed for implementation by practitioners, interdisciplinary scientists, and local leaders. Its robust findings, derived from a comparison between “synergistic” and “non-synergistic” cases, highlight critical factors for synergy creation, offering valuable insights for both theory and practice. The evidence base is strong, drawing on 17 cases of autonomous innovations across diverse geographical and cultural contexts, enhancing the generalizability of the conclusions.

The introduction, while detailed, could benefit from greater clarity and conciseness, focusing on synthesizing key concepts such as the role of synergies and leverage points. Some results, including the specific roles of different leverage points (LP(in), LP(out), LP(all)), would be more comprehensible with concrete examples from the analyzed cases. Although the article acknowledges the complexity of socio-ecological systems, it lacks a thorough discussion of methodological limitations, such as potential subjectivity in narrative analysis. While the article references a broad range of literature, a tighter integration of its findings with existing research on complex systems and socio-ecological transformations would be beneficial. The figures, particularly Figures 2 and 3, could be visually enhanced with clearer labels and more detailed explanations. Lastly, certain technical terms, such as “synergy-related value creation” could be better explained to improve accessibility for a non-specialist audience.

The article makes a significant contribution to the field of socio-ecological transformations and autonomous innovations. Enhancements in clarity, methodological transparency, and integration with existing literature would further strengthen its scientific and practical impact.

6. PLOS authors have the option to publish the peer review history of their article (what does this mean? ). If published, this will include your full peer review and any attached files.

**Do you want your identity to be public for this peer review?** For information about this choice, including consent withdrawal, please see our Privacy Policy .

Reviewer #1: No

Reviewer #2: No

---

## [Author Response · Author response to Decision Letter 0]

15 Feb 2025

To Reviewer #1

Thank you for the constructive, positive, and stimulating comments on our paper.

The first section of introduction (Transformation of social-ecological systems through synergies) was restructured into four paragraphs representing (1) to (4) and explanations were added in each paragraph to emphasize critical elements proposed by the reviewer #1.

The Introduction was divided into “Transformation of social-ecological systems through synergies,” “Research questions,” and “Theoretical framework” sections.

The Theoretical framework section was further divided into two subsections, “Network analysis for leverage point identification” and “Synergies to promote transformation of complex systems,” to highlight important concepts.

Table 1 was moved to the end of the first paragraph of methods.

We reorganized Materials and Methods into two sections covering initial data processing (Classification of autonomous innovations) and analytical procedures Analyses of processes and factors of synergy creation.

We made “Results,” “Discussion of Emerging Findings,” and “Conclusion, Limitations, and Future Research Agenda” sections. The theory and potential implications were covered in the “Conclusion, Limitations, and Future Research Agenda” section.

We moved fig 3 to the “Discussion of Emerging Findings” section and renumbered it as fig 5 to summarize the emerging findings on the processes and mechanisms of synergy creation in autonomous innovations.

Implications for societal actors and policymakers/development agencies were summarized at the end of the “Conclusion and guiding principles” subsection in the “Conclusion, Limitations, and Future Research Agenda” section.

We added discussions on the limitations of this study and current ideas to overcome these challenges in the “Limitations and Future Research Agenda” subsection.

We are strongly stimulated and motivated by your comments. We hope that we have made significant improvement in the clarity and logical consistency of the paper.

To Reviewer #2

We are delighted to receive positive and stimulating comments to improve our paper from the viewpoint of complex adaptive systems. Thank you very much for your feedback.

The Introduction was entirely reorganized to emphasize the key concepts in the theoretical framework subsection to achieve greater clarity.

Please also refer to the response to reviewer #1.

Examples were given in the methods at the part of explaining LP types using Fig 1A and 1B. Please refer to the paragraph from line 251 to 268.

We discussed potential subjectivity referring to our previous papers in the first paragraph of the “Analyses of processes and factors of synergy creation” subsection in “Materials and Methods.”.

The difficulties in avoiding arbitrariness and further research challenges were also discussed in detail in the “Limitations and Future Research Agenda” subsection at the end of the paper in reference to the transdisciplinary approach.

We added a paragraph at the beginning of the “Network analysis for leverage point identification” subsection in “Theoretical framework” to explain the relevance of this study in the complex adaptive system research.

We also discussed the importance of this study in the context of complex systems in the first paragraph of the“Discussion of Emerging Findings” section.

Eight new references were added in these paragraphs.

All network diagrams, including Figures 2, were revised to high-quality TIFF files to produce better visibility. Node labels were improved to make it clear. All figures were checked using PACE to meet PLOS requirements. Explanations were added in the text related to figure 2 at the second paragraph of the “Results” section.

We moved fig 3 to the “Discussion of Emerging Findings” section and renumbered it as fig 5. It was also checked using PACE. Explanations were added to summarize the emerging findings on the processes and mechanisms of synergy creation.

We carefully checked inadequate jargon and rewrote it for non-specialist audiences. For example, “synergy-related value creation” was rephrased to “creation of values leading to synergies.”

Thank you very much for your encouragement. We tried our best to strengthen the scientific and practical impacts of our paper.

---

## [Decision Letter · Decision Letter 1]

11 Mar 2025

PONE-D-24-53688R1Mechanisms of synergy creation for social-ecological transformation:  Leverage point analysis of the emergence of autonomous innovationsPLOS ONE

Dear Dr. Tajima,

Thank you for submitting your manuscript to PLOS ONE. After careful consideration, we feel that it has merit but does not fully meet PLOS ONE’s publication criteria as it currently stands. Therefore, we invite you to submit a revised version of the manuscript that addresses the points raised during the review process.

We look forward to receiving your revised manuscript.

Kind regards,

Enrico Ivaldi

Academic Editor

PLOS ONE

Journal Requirements:

Reviewers' comments:

Reviewer's Responses to Questions

**Comments to the Author**

1. If the authors have adequately addressed your comments raised in a previous round of review and you feel that this manuscript is now acceptable for publication, you may indicate that here to bypass the “Comments to the Author” section, enter your conflict of interest statement in the “Confidential to Editor” section, and submit your "Accept" recommendation.

Reviewer #1: All comments have been addressed

Reviewer #2: All comments have been addressed

2. Is the manuscript technically sound, and do the data support the conclusions?

Reviewer #1: Yes

Reviewer #2: Yes

3. Has the statistical analysis been performed appropriately and rigorously? 

Reviewer #1: Yes

Reviewer #2: Yes

4. Have the authors made all data underlying the findings in their manuscript fully available?

Reviewer #1: Yes

Reviewer #2: Yes

5. Is the manuscript presented in an intelligible fashion and written in standard English?

Reviewer #1: Yes

Reviewer #2: Yes

6. Review Comments to the Author

Reviewer #1: Dear Author(s),

Thank you for allowing me to review your work again. I believe that the revision process you implemented has improved the overall quality of the manuscript and its readability, greatly increasing its potential. However, in my opinion, there are still some aspects that need to be resolved. I hope that the following recommendations will improve and strengthen the manuscript:

• Introduction: I consider that the introductory section of the paper has been greatly improved, making some key aspects of the manuscript much more identifiable, such as the purpose of the article, which is now clearly identifiable. I recommend that the author(s) include some concrete examples of social ecological systems (SESs) to make the topic more accessible to the reader and reduce its abstractness. I would suggest giving some additional references for what is meant by ‘autonomous innovation’, even if they are then detailed further in the ‘Materials and Methods’ section, to give the reader an overview right from the start of the paper. I suggest the author(s) remove the subsection ‘Research Question’ and make the text all in a single section with the research questions inside, then move on to section 2, ‘Theoretical Framework’.

• Theoretical Framework: I appreciated the restructuring of the document, which made it much more readable and understandable, explaining the theoretical key of leverage points (LP) and the reference context that considers uncertainty and the achievement of the SDGs through potential synergies.

• Materials and Methods/Findings: I consider this section much improved in terms of structure and readability.

• Discussion: I would recommend that the author(s) call this section simply ‘Discussion’. I also recommend that the author(s) link the discussion of the results more closely to the theoretical lens adopted in the paper, specified in the ‘Theoretical Framework’, making it clear how the work fits in with existing theory, and citing some of the key references discussed in section 2.

• Conclusion, Limitations and Future Research Agenda: I consider the concluding section to be much improved.

I congratulate the author(s) on the extensive improvements made and hope that these additional comments will increase the potential and quality of the study. Good luck with your research!

Reviewer #2: Dear Authors,

Thanks for your revisions according to the comments. I am thankful for the effort you have put into enhancing the manuscript's clarity, structure, and overall quality. The reorganization of the Introduction, along with the addition of examples and supplementary references, truly helps to strengthen the theoretical foundation and make the paper easier to read for a broader readership.

Your diligent attempt to provide for potential subjectivity and to mention the challenges of avoiding arbitrariness adds depth along with transparency to your methodological approach. The changes to the figures, including the enhancement of

Overall, I am pleased with the revisions and believe that the manuscript is now publishable. I commend you for attempting to enhance the paper and address the reviewers' comments thoroughly.

I have no comments at this time and recommend the manuscript for acceptance.

7. PLOS authors have the option to publish the peer review history of their article (what does this mean? ). If published, this will include your full peer review and any attached files.

**Do you want your identity to be public for this peer review?** For information about this choice, including consent withdrawal, please see our Privacy Policy .

Reviewer #1: No

Reviewer #2: No

---

## [Author Response · Author response to Decision Letter 1]

27 Mar 2025

Ref. No.: PONE-D-24-53688R1

Title: Mechanisms of synergy creation for social-ecological transformation: Leverage point analysis of the emergence of autonomous innovations

PLOS ONE

Dear Editor,

Thank you for the opportunity to revise and resubmit the second revision of our manuscript to PLOS ONE. We are grateful to the two reviewers whose suggestions have allowed us to improve the manuscript to be publishable.

We made further revisions according to the comments. Please find below a point-by-point response to the referee's comments.

Reviewer #1: Dear Author(s),

Thank you for allowing me to review your work again. I believe that the revision process you

implemented has improved the overall quality of the manuscript and its readability, greatly

increasing its potential. However, in my opinion, there are still some aspects that need to be

resolved. I hope that the following recommendations will improve and strengthen the

manuscript:

Thank you very much for your positive and constructive comments on the manuscript. We try our best to improve the quality of the manuscript as indicated below following your valuable comments.

• Introduction: I consider that the introductory section of the paper has been greatly improved,

making some key aspects of the manuscript much more identifiable, such as the purpose of the

article, which is now clearly identifiable. I recommend that the author(s) include some

concrete examples of social ecological systems (SESs) to make the topic more accessible to

the reader and reduce its abstractness.

We briefly introduced examples of SESs (UNESCO Biosphere Reserves and Social-ecological Production Landscapes and Seascapes) in the introduction with three new references.

I would suggest giving some additional references for

what is meant by ‘autonomous innovation’, even if they are then detailed further in the

‘Materials and Methods’ section, to give the reader an overview right from the start of the

paper.

We added a sentence to explain the concept of autonomous innovation, referring to two papers.

I suggest the author(s) remove the subsection ‘Research Question’ and make the text all

in a single section with the research questions inside, then move on to section 2, ‘Theoretical

Framework’.

We removed the subsection ‘Research Question.’

• Theoretical Framework: I appreciated the restructuring of the document, which made it

much more readable and understandable, explaining the theoretical key of leverage points

(LP) and the reference context that considers uncertainty and the achievement of the SDGs

through potential synergies.

Thank you very much for your kind comments.

• Materials and Methods/Findings: I consider this section much improved in terms of structure

and readability.

We are pleased to know the improved readability of these sections.

• Discussion: I would recommend that the author(s) call this section simply ‘Discussion’.

The title of this section was changed to ‘Discussion’.

I also recommend that the author(s) link the discussion of the results more closely to the

theoretical lens adopted in the paper, specified in the ‘Theoretical Framework’, making it clear

how the work fits in with existing theory, and citing some of the key references discussed in

section 2.

We added or modified five sentences in ‘Discussion’ to link our arguments with the theoretical lens proposed in ‘Theoretical Framework.’ Five key references were cited to show the relevance of our arguments with existing theory.

• Conclusion, Limitations and Future Research Agenda: I consider the concluding section to

be much improved.

Thank you very much. We also added another paragraph in ‘Limitation and Future Research Agenda’ to address future research opportunities on synergy creation mechanisms across much broader societal sectors such as education, health, and social welfare. Two new references were added with this regard.

I congratulate the author(s) on the extensive improvements made and hope that these

additional comments will increase the potential and quality of the study. Good luck with your

research!

We would like to express our deep gratitude for receiving your valuable comments throughout the review process. It was an important learning opportunity for us to improve our research.

Reviewer #2: Dear Authors,

Thanks for your revisions according to the comments. I am thankful for the effort you have

put into enhancing the manuscript's clarity, structure, and overall quality. The reorganization

of the Introduction, along with the addition of examples and supplementary references, truly

helps to strengthen the theoretical foundation and make the paper easier to read for a broader

readership.

Your diligent attempt to provide for potential subjectivity and to mention the challenges of

avoiding arbitrariness adds depth along with transparency to your methodological approach.

The changes to the figures, including the enhancement of

Overall, I am pleased with the revisions and believe that the manuscript is now publishable. I

commend you for attempting to enhance the paper and address the reviewers' comments

thoroughly.

I have no comments at this time and recommend the manuscript for acceptance.

We are pleased to receive your in-depth understanding of the efforts we have made to improve the manuscript according to your comments. It was a great opportunity for us to deepen our understanding of the complex adaptive systems and transparency in narrative analyses. We are really indebted to your valuable thoughts and comments.

---

## [Decision Letter · Decision Letter 2]

9 Apr 2025

Mechanisms of synergy creation for social-ecological transformation:  Leverage point analysis of the emergence of autonomous innovations

PONE-D-24-53688R2

Dear Dr. Tajima,

We’re pleased to inform you that your manuscript has been judged scientifically suitable for publication and will be formally accepted for publication once it meets all outstanding technical requirements.

Kind regards,

Enrico Ivaldi

Academic Editor

PLOS ONE

Additional Editor Comments (optional):

The reviewers' recommendations have been followed: I wish the authors good continuation with their research

Reviewers' comments:

Reviewer's Responses to Questions

**Comments to the Author**

1. If the authors have adequately addressed your comments raised in a previous round of review and you feel that this manuscript is now acceptable for publication, you may indicate that here to bypass the “Comments to the Author” section, enter your conflict of interest statement in the “Confidential to Editor” section, and submit your "Accept" recommendation.

Reviewer #1: All comments have been addressed

Reviewer #2: All comments have been addressed

2. Is the manuscript technically sound, and do the data support the conclusions?

Reviewer #1: Yes

Reviewer #2: Yes

3. Has the statistical analysis been performed appropriately and rigorously? 

Reviewer #1: Yes

Reviewer #2: Yes

4. Have the authors made all data underlying the findings in their manuscript fully available?

Reviewer #1: Yes

Reviewer #2: Yes

5. Is the manuscript presented in an intelligible fashion and written in standard English?

Reviewer #1: Yes

Reviewer #2: Yes

6. Review Comments to the Author

Reviewer #1: (No Response)

Reviewer #2: (No Response)

7. PLOS authors have the option to publish the peer review history of their article (what does this mean? ). If published, this will include your full peer review and any attached files.

**Do you want your identity to be public for this peer review?** For information about this choice, including consent withdrawal, please see our Privacy Policy .

Reviewer #1: No

Reviewer #2: No

---

## [Editor Report · Acceptance letter]

PONE-D-24-53688R2

PLOS ONE

Dear Dr. Tajima,

I'm pleased to inform you that your manuscript has been deemed suitable for publication in PLOS ONE. Congratulations! Your manuscript is now being handed over to our production team.

Kind regards,

on behalf of

Prof. Enrico Ivaldi

Academic Editor

PLOS ONE